# The Relationship Between Patient Activation, Cancer-Related Symptoms, and Job Performance Among Young Adult Cancer Survivors with Low and High Workplace Support: A Cross-Sectional Observational Study

**DOI:** 10.3390/cancers17111742

**Published:** 2025-05-22

**Authors:** Takafumi Soejima, Masahito Tokita, Mari Kitao

**Affiliations:** 1Graduate School of Health Sciences, Kobe University, 7-10-2 Tomogaoka, Suma-ku, Kobe 654-0142, Hyogo, Japan; 2Keio Research Institute at SFC, Keio University, 5322 Endo, Fujisawa 252-0882, Kanagawa, Japan

**Keywords:** cancer survivors, long-term adverse effects, patient participation, social support, work performance, young adults

## Abstract

Optimal work outcomes were crucial for young adult cancer survivors due to the formative years of their careers. Maintaining job performance and pursuing employment among young adult cancer survivors reduced financial burden and obtained a sense of normality, thereby improving health-related quality of life. However, young adult cancer survivors would be at high risk of poor job performance due to cancer-related symptoms. Our study provides a deeper understanding of the protective role of workplace support and patient activation for the adverse effects of cancer-related symptoms on job performance among young adult cancer survivors. This study’s findings suggest that healthcare staff should first assess the workplace support received by young adult cancer survivors, and if the workplace support is insufficient, healthcare staff can enhance the survivors’ self-management skills to improve job performance.

## 1. Introduction

Maintaining job performance and pursuing employment after a cancer diagnosis have been demonstrated to reduce financial burden, enhance self-esteem, and obtain a sense of normality, thereby improving health-related quality of life among cancer survivors [1,2,3]. Cancer survivors were at higher risk of poor job performance than individuals without a cancer history [4,5]. Moreover, some cancer survivors often experienced physical fatigue, depression, and cognitive impairment, which were related to cancer and its therapy and persisted for years after treatment completion [6,7,8]. These symptoms were also risk factors for impaired job performance [9,10,11,12,13]. Among young adult cancer survivors, who are commonly defined as individuals aged 20–39 who were diagnosed with cancer in adulthood and have completed treatment [14], a further understanding of the work outcomes and these related factors will be essential for the future development of relevant services and resources for optimal work conditions.

In Japan, the Act for Eliminating Discrimination against Persons with Disabilities mandates that companies provide reasonable accommodation in the workplace for individuals holding a disability certificate [15]. Additionally, under the Act to Facilitate the Employment of Persons with Disabilities [16], individuals with a disability certificate can utilize the employment quota held only for persons with disabilities and receive job support coaching. Although some cancer survivors had a disability certificate due to cancer and its treatment-related complications and were able to utilize these resources, many did not possess a disability certificate [17]. When a cancer survivor does not have a disability certificate, there are few public social resources available for them to utilize. Based on the guidelines for supporting the employment of cancer survivors [18], multidisciplinary teams at hospitals (e.g., healthcare staff, labor and social security attorney, social workers), Hello Work (public employment service center in Japan), and companies provide support to cancer survivors.

Workplace support plays an important role in improving job performance among cancer survivors, and the provision of job accommodation had a positive impact on return-to-work experience and job performance in cancer survivors or chronically ill individuals [19,20,21]. Employed cancer survivors often disclosed their cancer history at work and gained workplace support [22,23]. However, some reported workplace discrimination, including bullying and refusing accommodations, as well as constrained opportunities for career development [24]. Japanese young adult cancer survivors are also more likely to experience discrimination and prejudice related to cancer in their workplaces than older cancer survivors [25]. Furthermore, 35% of Japanese cancer survivors quit their jobs because they do not want to be a burden on the workplace, suggesting that it is difficult to request support for themselves from the workplace [26]. This situation indicated that young adult cancer survivors may not always receive workplace support. When obtaining workplace support is difficult and ineffective to improve work outcomes among cancer survivors, interventions targeting individual characteristics should be prioritized [27]. However, few studies have assessed modifiable individual factors, such as one’s problem-solving style (i.e., behavioral and cognitive traits in the face of problems) [27].

Patient activation is considered an important individual characteristic for improving job performance among cancer survivors. It is defined as an individual’s knowledge, skills, confidence, and behaviors for managing their health [28]. Patient activation relates to the act of making changes in health behaviors and a reduction in symptoms such as fatigue and psychological distress, resulting in better health-related quality of life among individuals with chronic conditions [29,30,31]. Patient activation also improved job performance in terms of working accurately, finishing work on time, and concentrating on the job among cancer survivors [32]. Given that cancer-related symptoms were associated with low job performance [9,10,11,12,13] and high patient activation was associated with a reduction in cancer-related symptoms [29], activation of self-management may be an indirect cause for improved job performance through cancer-related symptoms. However, there was a paucity of empirical evidence on the direct and indirect effects of patient activation on job performance.

It is possible that patient activation is more valuable for maintaining job performance among young adult cancer survivors with low workplace support than among those with high workplace support. In a supportive work environment, cancer survivors can work according to their symptoms by adjusting the job demands, changing their tasks, and using flexible work arrangements, thereby mitigating the negative effects of their symptoms on their job [33,34,35]. Ten percent of Japanese cancer survivors quit their jobs due to difficulties in getting support from their workplace [26]. Additionally, 5–15% of Japanese cancer survivors leave their jobs because they are physically and psychologically unable to work [26], and managing their symptoms is also necessary for young adult cancer survivors to improve work outcomes. When young adult cancer survivors cannot mitigate the negative effects of their symptoms on their work by workplace support, they may need to reduce their negative impact on their work by handling their symptoms on their own; that is, patient activation would be more crucial for young adult cancer survivors without workplace support to maintain job performance than with workplace support. Japanese workers who received less support from their supervisors and coworkers were more likely to manage and improve their health on their own than those who received more support from their supervisors and coworkers [36]. In this way, the impact of patient activation on cancer-related symptoms and job performance among young adult cancer survivors may vary depending on workplace support. However, no previous studies have examined this difference due to workplace support among young adult cancer survivors.

Our study thus aimed to identify differences in the relationship between patient activation, cancer-related symptoms, and job performance among young adult cancer survivors with low and high workplace support. We tested this hypothesis to suggest and devise effective interventions that would improve job performance among young adult cancer survivors, considering workplace support. Furthermore, clarifying whether patient activation directly or indirectly affects job performance would help determine whether to prioritize interventions on patient activation or cancer-related symptoms.

## 2. Materials and Methods

### 2.1. Study Participants and Procedures

We employed a cross-sectional observational study utilizing a web-based survey. Inclusion criteria for participants were as follows: (1) They were 20–39 years old at the time of survey; (2) they were diagnosed with cancer at an age of 20 years or older; (3) they were employed; (4) they completed treatment for cancer; and (5) they could understand the purpose of this study as well as complete a questionnaire in Japanese.

In January 2022, we invited 6161 young adult cancer survivors aged 20–39 years who had registered with the commercial cancer panel system of Macromill Inc. (https://group.macromill.com, accessed on 18 August 2021) to participate in this study via email. Macromill Inc. is an internet research company based in Japan. Participants who were interested in this study logged into the online survey system provided by Macromill Inc. on their own devices (tablets, smartphones, etc.). Of 6161 young adult cancer survivors, 693 returned the online questionnaires (response rate, 11.3%). This response rate was similar to that in a previous study [37]. Participants who were diagnosed with cancer at less than 20 years of age (*n* = 138), were unemployed (*n* = 120), did not complete treatment (*n* = 216), and did not complete the WHO-HPQ (*n* = 17) were excluded to investigate job performance in employed survivors. Finally, we analyzed data from 202 young adult cancer survivors.

They were informed about this study and gave researchers their consent to participate in the online survey system. They answered the questionnaire only if they gave their consent to participate. This study was approved by the Ethics Committee of the principal investigator’s institution (No. 1029).

### 2.2. Measures

#### 2.2.1. Demographic and Clinical Characteristics

The following demographic and clinical characteristics were obtained from the participants through the questionnaire: age, gender, educational level, cancer sites, time since diagnosis, treatment, type of employment and occupation, job demands, and workplace support. We used two subscales of the New Brief Job Stress Questionnaire (New BJSQ) to assess job demands and workplace support, and each subscale had six items [38]. Each subscale score is calculated as the sum of item scores divided by the number of items. The missing item scores were imputed with the mean score of other items of the same subscale. If more than half of the items have missing responses, the score is not calculated. The possible response range for each subscale score is 1 to 4, and a higher score indicates more job demands or workplace support. The Cronbach’s alpha in this study was 0.86 for the subscale on job demands and 0.93 for the subscale on workplace support. The national average score of the workplace support subscale of the New BJSQ was 2.53 points in Japan [38]. Thus, participants who scored less than 2.53 points on the workplace support subscale of the New BJSQ were categorized into the “low support group”, and those who scored 2.53 points or higher were categorized into the “high support group”.

#### 2.2.2. Patient Activation

Patient activation was evaluated by the Japanese version of the Patient Activation Measure-13 (PAM13) [39], which is composed of 13 items using a four-point Likert scale on the respondents’ knowledge, skills, confidence, and behaviors for managing their own health. The patient activation score is derived by converting the sum of all item scores to a range of 0–100. The missing values of item scores were imputed using the mean score of the completed items on the PAM13. If more than half of the items are missing, the patient activation score is not calculated. A higher patient activation score indicated a more proactive involvement in self-management. The Japanese version of the PAM13 was a validated and reliable instrument for patients with mental disorders in the internal consistency, test–retest reliability, factorial validity using the Rasch analysis, and concurrent validity against self-efficacy for treatment adherence [39]. The Cronbach’s alpha was 0.87 in the present study.

#### 2.2.3. Cancer-Related Symptoms

Physical fatigue was measured using the Physical Fatigue subscale of the Cancer Fatigue Scale (CFS) [40]. The CFS includes physical, affective, and cognitive fatigue subscales; however, the present study used only the Physical Fatigue subscale because the other subscales measure psychological and cognitive fatigue. The Physical Fatigue subscale consists of seven items rated on a five-point Likert scale, and the subscale score is the sum of these item scores. The missing values of item scores were replaced with the mean score of completed items. If more than half of the items are missing, the score is not calculated. The subscale score ranges from 0 to 28, with a higher score indicating more severe physical fatigue. The Cronbach’s alpha was 0.94 in the present study.

Depressive symptoms were evaluated using the Kessler-6 (K6) [41], which is composed of six items with a five-point Likert scale. The K6 score is the sum of all item scores. The missing values of item scores were imputed using the mean score of completed items of the K6. If more than half of the items are missing, the K6 score is not computed. The score ranges from 0 to 24, indicating that a high score indicates severe depressive symptoms. The Cronbach’s alpha was 0.94 in the present study.

The Cognitive Functioning subscale of the European Organization for Research and Treatment of Cancer Quality of Life Questionnaire Core 30 (EORTC QLQ-C30) was used to measure the participants’ cognitive impairments [42]. This subscale includes two items and uses a four-point Likert scale. The subscale score is calculated by transforming the average score for each item answered between 0 and 100. The missing item scores were imputed with the mean score of other items of the same subscale. If all the items have missing responses, the score is not calculated. A higher score indicates better cognitive functioning. This subscale has been used in numerous studies to assess the cognitive impairment of cancer survivors [43]. The Cronbach’s alpha in this study was 0.79.

#### 2.2.4. Job Performance

We used the short form of the World Health Organization Health and Performance Questionnaire (WHO-HPQ) to assess job performance [44]. The WHO-HPQ measures the respondents’ actual performance using one numerical rating item from 0 to 10. This score is calculated as the item score multiplied by 10 and ranges from 0 (total lack of job performance) to 100 (no lack of job performance). If the single item of the WHO-HPQ is not responded to, the score is not calculated. A higher score on the respondents’ actual performance indicates better job performance.

### 2.3. Statistical Analysis

Frequencies and percentages for categorical variables and median with interquartile range (IQR) for continuous variables were calculated. Differences in categorical and continuous variables between the low and high support groups were assessed by the chi-squared test and the Mann–Whitney U test, respectively. Furthermore, for the WHO-HPQ, correlation ratios (η^2^) with categorical variables and Spearman’s rank correlation coefficients (r) with continuous variables were calculated. The demographic and clinical characteristics with significant differences between low and high support groups or significantly associated with WHO-HPQ were identified as potential confounders for the association of job performance with patient activation and cancer-related symptoms.

We hypothesized that, only among young adult cancer survivors with low workplace support, patient activation would be directly related to job performance, as well as indirectly related to job performance through cancer-related symptoms (Figure 1). Multiple-group structural equation modeling (SEM) was performed to examine differences in the relationship between patient activation, cancer-related symptoms, and job performance between young adult cancer survivors with low and high workplace support. First, we developed the model on the relationship between patient activation, cancer-related symptoms, and job performance. Cancer-related symptoms were set as a latent variable consisting of physical fatigue, depression, and cognitive impairments. Second, we compared the model including only the main variables (i.e., patient activation, cancer-related symptoms, and job performance) with the model including the main and adjusted variables with significant differences between the low and high support groups, using the indicators of model fit. Type of employment (1 = Temporary, 0 = Others), cancer sites (1 = Female genital cancer, 0 = Others), and period from diagnosis (1 = 5 years or more, 0 = Others) were entered as adjusted variables into the model. In the model including the main and adjusted variables, there was a significant association between the type of employment and patient activation, but no other significant association between the main and adjusted variables. Thus, the model including the main variables and type of employment with the path from type of employment and patient activation was created. The model including only the main variables (CMIN/df = 0.39, CFI = 1.00, GFI = 0.99, AGFI = 0.98, and RMSEA = 0.00) indicated better fit than the model including the main variables and type of employment (CMIN/df = 1.25, CFI = 0.97, GFI = 0.94, AGFI = 0.90, and RMSEA = 0.04). We therefore used the multiple-group SEM model, including only patient activation, cancer-related symptoms, and job performance, in further analyses. Third, the model fit indicators were compared between the model without any path constraints and the models with equality constraints of each path and all paths to assess whether each relationship between the main variables was moderated depending on workplace support. Significant changes in chi-squared statistics in the constrained model indicated that path coefficients could be different between the low and high support groups. Insignificant changes in chi-squared statistics in the constrained model indicated that path coefficients could be equal. Finally, in the sensitivity analysis, we compared eight models, including all possible paths between patient activation, cancer-related symptoms, and job performance, using model fit indices. A good model fit was indicated by chi-square statistics divided by degrees of freedom (CMIN/df) < 2, comparative fit index (CFI) > 0.97, goodness-of-fit index (GFI) > 0.95, adjusted goodness-of-fit index (AGFI) > 0.90, and root mean square error of approximation (RMSEA) < 0.05 [45]. Standardized path coefficients (β) with 95% confidence intervals (CIs) were estimated using a bootstrap analysis with 2000 iterations and were tested by the bias-corrected bootstrap test to robustly estimate parameters in structural equation modeling with variables that are not normally distributed.

All analyses were performed using IBM SPSS and Amos software version 24 (SPSS, Inc., Chicago, IL, USA), and the significance level was set at 0.05 (two-tailed).

## 3. Results

The low and high support groups consisted of 97 (48%) and 105 (52%) participants, respectively (Table 1). Of the participants, 107 (53%) were aged 35–39 years, 157 (78%) were female, and 135 (67%) were permanent workers. The most common diagnosis was female genital cancer (*n* = 85, 42%), including uterine (*n* = 76, 38%) and ovarian cancer (*n* = 9, 4%). A total of 122 participants (60%) received surgery. Overall, 72 (36%) of the participants were diagnosed more than five years ago. The high support group had a higher percentage of permanent workers (*p* = 0.01), a lower percentage of female genital cancer (*p* = 0.04), and a shorter time since diagnosis (*p* = 0.02) than did the low support group. No correlation coefficients between characteristics and job performance were significant. The median scores of PAM13 and WHO-HPQ were 51.0 (IQR = 45.3–63.1) and 60.0 (IQR = 50.0–80.0), respectively. The high support group reported a higher score on the WHO-HPQ than did the low support group (*p* < 0.01).

The multiple-group SEM showed a good model fit: CMIN/df = 0.389, CFI = 1.000, GFI = 0.994, AGFI = 0.977, and RMSEA < 0.001 (Figure 2). In the model of the low support group, the path coefficients from patient activation to cancer-related symptoms (β = −0.30, 95% CI −0.46 to −0.05, *p* = 0.01), cancer-related symptoms to job performance (β = −0.29, 95% CI −0.54 to −0.06, *p* = 0.01), and patient activation to job performance (β = 0.28, 95% CI 0.08 to 0.48, *p* < 0.01) were significant. On the other hand, in the model of the high support group, the path coefficients from patient activation to cancer-related symptoms (β = −0.06, 95% CI −0.31 to 0.26, *p* = 0.76), cancer-related symptoms to job performance (β = −0.12, 95% CI −0.42 to 0.15, *p* = 0.37), and patient activation to job performance (β = 0.20, 95% CI −0.08 to 0.43, *p* = 0.21) were not significant.

In the comparisons of the model, the change in the chi-square statistics between the model without any path constraints and the other models was not statistically significant (Table 2). However, the absolute indicators (e.g., GFI and AGFI) for the unconstrained model indicated a better fit than that for the model constraining each path and all paths.

## 4. Discussion

This study demonstrated that, only among young adult cancer survivors with low workplace support, patient activation was directly related to job performance and indirectly related to job performance through cancer-related symptoms, which supported our hypothesis. Furthermore, the differences in path coefficients between patient activation, cancer-related symptoms, and job performance between low and high support groups would be supported by the changes in the model fit index (e.g., GFI and AGFI), while there was no significant difference in path coefficients between the low and high support groups. However, it may be premature to conclude the moderating effect of workplace support in the relationship between patient activation, cancer-related symptoms, and job performance. Even if employed cancer survivors often disclosed their cancer history at work [22,23], some reported workplace discrimination, as well as low workplace support [24]. Of Japanese adult cancer survivors, 10% reported that it was difficult for them to receive enough support from their workplace [26]. Our findings on the differences in the impact of patient activation on cancer-related symptoms and job performance depending on workplace support should be considered because of the practical importance of the role of patient activation among Japanese young adult cancer survivors lacking workplace support.

Our study demonstrated that the direct and positive relationship between patient activation and job performance was observed only for young adult cancer survivors with low workplace support. In a previous study, cancer survivors with high patient activation were less likely to feel stressed at their job, which led to working accurately and finishing their job with concentration [32]. Patient activation reflects the sense of control in not only the health-related aspect but also other life domains such as work [32,46]. Employees’ inability to control physical and psychological burdens in a job leads to lower productivity, increased sickness absence, and poor health and well-being [47]. Young adult cancer survivors with high patient activation may have a sense of controlling physical and psychological burden at their job, which results in improved job performance.

On the contrary, our study found that patient activation had no significant impact on job performance among young adult cancer survivors with high workplace support. The characteristics of any job are divided into job demands and resources [48]. Job resources stimulate employee initiatives to complete work goals [49]. Past studies showed that workplace support as a job resource resulted in shared values and attitudes of trust and reciprocity, as well as practices of collective action in a workplace, such as assisting other employees in completing work-related duties, which increased job performance [48,50]. This supportive work environment would enable young adult cancer survivors to maintain job performance, regardless of patient activation.

Previous studies did not find the mediating role of health conditions for the relationship between patient activation and work outcomes, including participation in paid work and work-related problems [32]. Our finding that patient activation was indirectly associated with job performance through cancer-related symptoms among young adult cancer survivors with low workplace support was notable. In addition, although numerous previous studies reported modifiable workplace factors (e.g., physical and psychological job demands) that were relevant to job performance, few studies identified modifiable individual factors [27]. Our result highlights the importance of patient activation as an individual factor for improving job performance in a situation where workplace support and accommodation are less available.

On the other hand, our study did not indicate an indirect relationship between patient activation and job performance among young adult cancer survivors with high workplace support. Furthermore, this result suggests that workplace support reduces the adverse effect of cancer-related symptoms on job performance. One possible reason for this is that a supportive work environment does not require young adult cancer survivors to handle cancer-related symptoms and maintain job performance themselves through self-management. Among cancer survivors who received enough workplace support, supervisors and coworkers adjusted survivors’ burdens and tasks at the job according to their symptoms and promoted their utilization of remote work and flexible timings [34,35]. Cancer survivors who availed themselves of a flexible work arrangement also maintained a balance between working and taking breaks, depending on their condition, without hesitation to coworkers and supervisors [33]. This workplace accommodation would enable young adult cancer survivors to work depending on their cancer-related symptoms, reducing their adverse effects on job performance and maintaining job performance, regardless of their self-management.

The results of this study suggested that workplace support reduced the adverse effects of cancer-related symptoms on job performance, though workplace support reduced the relationship between patient activation and job performance. A previous study and a guideline indicated that collaboration between healthcare staff and workplaces was important while making workplace accommodations for young adult cancer survivors [18,26]. Firstly, wherein nurses and other healthcare staff collect information from young adult cancer survivors about their job, and upon receiving survivors’ consent, healthcare staff and workplaces (i.e., employers, supervisors, coworkers, occupational physicians, or occupational health nurses) need to share their cancer diagnosis, treatment modalities, possible or existing cancer-related complications, which negatively affect job performance, job tasks they are required to complete, and accommodations needed in their workplaces. Labor and social security attorneys or social workers may need to advise workplaces about laws and social welfare systems regarding workplace support and accommodations for young adult cancer survivors. The Basic Plan to Promote Cancer Control Programs in Japan prescribes that employers should support cancer survivors in balancing treatment and follow-up with working, promoting their understanding at the workplace, and forming a work climate that is suitable for them [51]. When healthcare staff share information about young adult cancer survivors with their workplace, they would be more likely to receive workplace support. Even if young adult cancer survivors do not want to disclose such information to then receive enough support from their workplace, our findings suggested that nurses and other healthcare staff need to assess the existing symptoms of each young adult cancer survivor and provide them with tailored advice on symptom management and lifestyle change using relevant materials and websites, in order to enhance their patient activation. Healthcare staff should also assess whether young adult cancer survivors are confident in managing their symptoms and successfully implement these management strategies after advising them on symptom management. Some previous studies showed that a web-based intervention related to patient activation had a positive effect on patient activation in young adult cancer survivors than in older cancer survivors [52,53]. A similar web-based intervention may be more accessible and valuable for young adult cancer survivors, who are familiar with tablets and smartphones; this may help promote patient activation and manage their health conditions, thereby improving job performance.

Our study had certain limitations. First, the response rate of our study was low. The present study also included a high proportion of female genital cancer survivors, which reflected the proportion of cancer types among young adults in Japan [54]. In addition, many of the participants had received surgery and may have had less severe complications than those who had received chemotherapy or radiation therapy. Thus, our study participants may be biased toward younger adult cancer survivors in better health conditions, and our findings had limited generalizability. Second, the participants had a higher educational level, considering that 54% graduated from university or graduate school, compared to the general Japanese population [55]. Previous studies reported that educational level was positively related to patient activation and job performance among cancer survivors [32,33]. Our study may overestimate the relationship between patient activation and job performance. Third, while the WHO-HPQ is a scale with sufficient validity and reliability, this scale is a single-item scale and may not adequately measure job performance. Further research should use a multidimensional measure of job performance, such as the Work Limitations Questionnaire [56], to verify our findings. Fourth, this study supported our hypothesis by the multiple-group SEM, indicating a good model fit, but it had a cross-sectional observational design. Thus, causal relationships between patient activation, cancer-related symptoms, and job performance should be interpreted with caution. We set the path from patient activation to cancer-related symptoms on the basis of previous studies [30,31,32], but the SEM was not able to confirm whether the arrow from patient activation to cancer-related symptoms or from cancer-related symptoms to patient activation was appropriate. Further study should clarify the temporality of relationships between patient activation, cancer-related symptoms, and job performance using a longitudinal design to identify this causal association. Finally, our study did not clarify the detailed workplace support that young adult cancer survivors received. Previous studies reported that cancer survivors received various types of workplace support, such as remote work and flextime [33,34,35]. Future research should inspect the differences in our findings by type of workplace support in effectively maintaining job performance.

## 5. Conclusions

The results indicated that patient activation would be directly related to job performance and be indirectly related to it through cancer-related symptoms only among young adult cancer survivors with low workplace support. Our findings suggest the importance of workplace support for young adult cancer survivors and patient activation for those with low workplace support in maintaining job performance. However, this study did not confirm the causal relationship between patient activation, cancer-related symptoms, and job performance due to a cross-sectional observational design. Given the limitations of the present study design, our findings require further research.

## Figures and Tables

**Figure 1 cancers-17-01742-f001:**
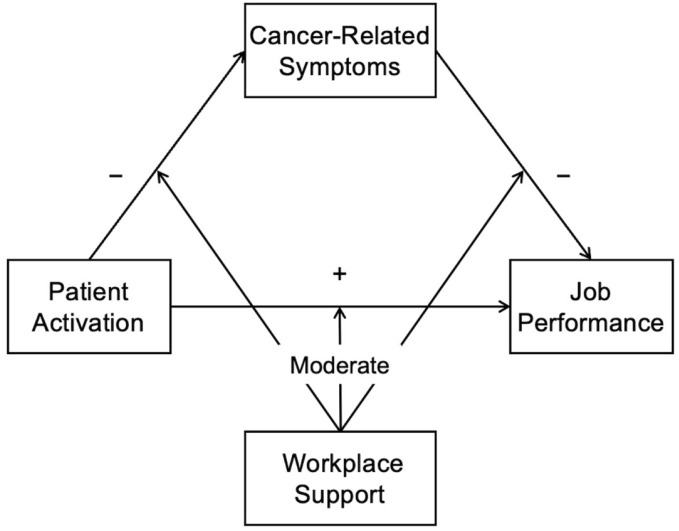
Hypothesis model of this study. Note. The symbol “+” indicates a positive relationship, and the symbol “−” indicates a negative relationship. Workplace support moderates the relationship between patient activation and cancer-related symptoms, between patient activation and job performance, and between cancer-related symptoms and job performance.

**Figure 2 cancers-17-01742-f002:**
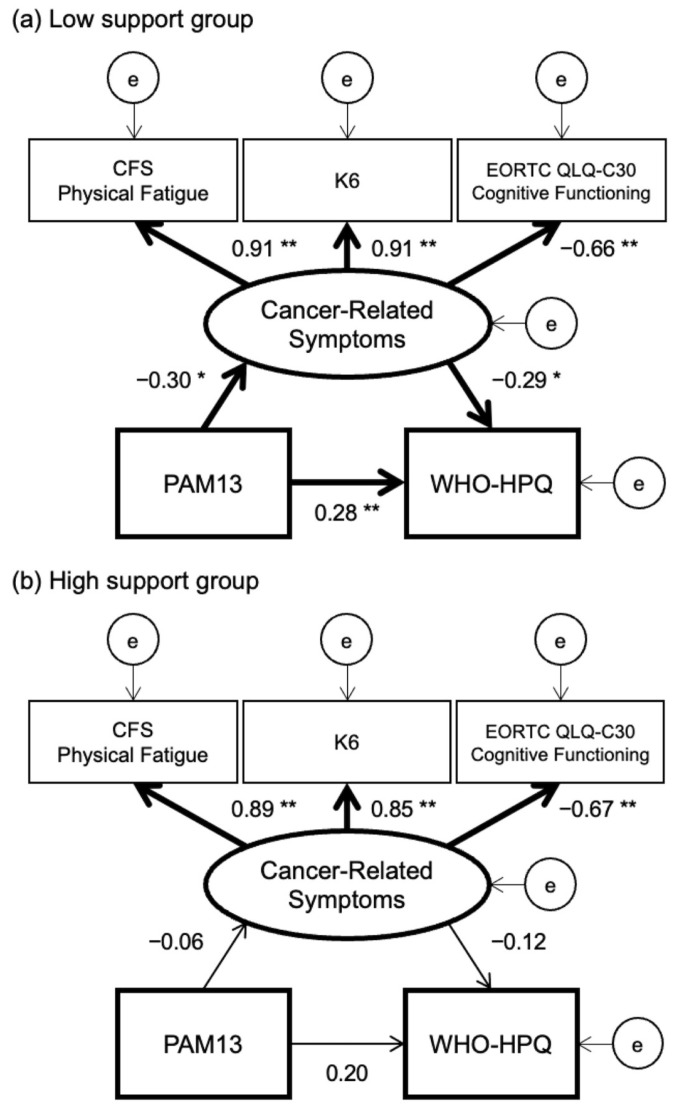
Multiple-group SEM of the relationship between patient activation, cancer-related symptoms, and job performance. Note. The results of multiple-group SEM in (**a**) the low support group and (**b**) the high support group are shown. The latent variable “e” implies the error of each variable. Significant paths are in bold. * *p* < 0.05. ** *p* < 0.01.

**Table 1 cancers-17-01742-t001:** Characteristics of participants.

*N* = 202											
	Total		Low Support Group (*n* = 97)		High Support Group (*n* = 105)			WHO-HPQ			
	*n*	%	*n*	%	*n*	%	*p* ^a^	Median	IQR	η^2 b^	*p* ^b^
Age							0.07			0.14	0.29
20–24 years	7	3	2	2	5	5		50.0	30.0–80.0		
25–29 years	27	13	19	20	8	8		60.0	50.0–80.0		
30–34 years	61	30	28	29	33	31		70.0	50.0–80.0		
35–39 years	107	53	48	49	59	56		60.0	50.0–70.0		
Gender							0.06			0.06	0.40
Male	45	22	16	16	29	28		70.0	50.0–80.0		
Female	157	78	81	84	76	72		60.0	50.0–80.0		
Educational level							0.89			0.19	0.11
Junior-high school	4	2	2	2	2	2		65.0	52.5–70.0		
High school	43	21	20	21	23	22		60.0	50.0–70.0		
College/Vocational school	42	21	18	19	24	23		60.0	50.0–70.0		
University/Graduate school	110	54	56	58	54	51		70.0	50.0–80.0		
Others	3	1	1	1	2	2		60.0	50.0–60.0		
Type of employment							0.01			0.07	0.66
Permanent	135	67	55	57	80	76		60.0	50.0–80.0		
Temporary	58	29	37	38	21	20		60.0	50.0–72.5		
Self-employed	9	4	5	5	4	4		70.0	60.0–75.0		
Type of occupation							0.80			0.18	0.37
Technical	38	19	17	18	21	20		60.0	50.0–80.0		
Managerial	17	8	8	8	9	9		70.0	60.0–80.0		
Clerical	76	38	35	36	41	39		70.0	50.0–80.0		
Sales	20	10	9	9	11	10		60.0	50.0–70.0		
Production	16	8	7	7	9	9		70.0	52.5–80.0		
Services	25	12	14	14	11	10		60.0	50.0–70.0		
Others	10	5	7	7	3	3		60.0	50.0–75.0		
Cancer sites							0.04			0.24	0.07
Gastric cancer	16	8	6	6	10	10		70.0	52.5–80.0		
Colorectal cancer	16	8	8	8	8	8		65.0	50.0–77.5		
Breast cancer	18	9	11	11	7	7		70.0	60.0–72.5		
Female genital cancer	85	42	50	52	35	33		60.0	50.0–80.0		
Thyroid cancer	17	8	7	7	10	10		50.0	50.0–70.0		
Lymphoma/leukemia	15	7	4	4	11	10		70.0	50.0–80.0		
Others	35	17	11	11	24	23		70.0	40.0–70.0		
Period from diagnosis							0.02			0.11	0.27
<1 year	32	16	15	15	17	16		70.0	40.0–80.0		
1– <5 years	98	49	38	39	60	57		60.0	50.0–80.0		
≥5 years	72	36	44	45	28	27		70.0	50.0–77.5		
Surgery							0.19			0.00	0.97
Yes	122	60	54	56	68	65		60.0	50.0–80.0		
No	80	40	43	44	37	35		60.0	50.0–72.5		
Chemotherapy							0.51			0.02	0.76
Yes	50	25	22	23	28	27		60.0	50.0–80.0		
No	152	75	75	77	77	73		70.0	50.0–80.0		
Radiation							0.14			0.01	0.87
Yes	26	13	9	9	17	16		60.0	50.0–80.0		
No	176	87	88	91	88	84		60.0	50.0–70.0		
Hormone therapy							0.50			0.05	0.44
Yes	18	9	10	10	8	8		60.0	50.0–80.0		
No	184	91	87	90	97	92		70.0	60.0–80.0		
	**Median**	**IQR**	**Median**	**IQR**	**Median**	**IQR**	***p* ^a^**	**Median**	**IQR**	**r ^c^**	***p* ^c^**
Job demands on New BJSQ	2.7	2.2–3.2	2.5	2.0–3.1	2.7	2.2–3.2	0.18	-	-	−0.08	0.25
Workplace support on New BJSQ	2.3	2.0–2.8	2.0	1.5–2.8	2.3	2.0–2.9	0.03	-	-	0.27	<0.01
Patient activation score on PAM13	51.0	45.3–63.1	51.0	45.3–60.6	53.2	45.3–63.1	0.58	-	-	0.31	<0.01
CFS Physical Fatigue subscale	7.0	3.0–13.3	7.0	3.0–13.5	7.0	2.5–13.5	0.99	-	-	−0.16	0.02
K6	5.0	1.0–11.0	6.0	0.0–10.5	4.0	1.0–11.5	0.71	-	-	−0.16	0.02
EORTC QLQ-C30 Cognitive Functioning subscale	66.7	50.0–83.3	66.7	50.0–100.0	66.7	50.0–83.3	0.37	-	-	0.18	<0.01
WHO-HPQ	60.0	50.0–80.0	60.0	50.0–70.0	70.0	60.0–80.0	<0.01	-	-	-	-

Note. CFS, Cancer Fatigue Scale; EORTC QLQ-C30, European Organization for Research and Treatment of Cancer Quality of Life Questionnaire Core 30; IQR, interquartile range; K6, Kessler-6; New BJSQ, New Brief Job Stress Questionnaire; PAM13, Patient Activation Measure-13; WHO-HPQ, World Health Organization Health and Performance Questionnaire. ^a^ The differences in characteristics between groups were tested by the chi-squared test for categorical variables and the Mann–Whitney U test for continuous variables. ^b^ The correlation ratio between characteristics and WHO-HPQ and *p*-values were calculated. ^c^ The Spearman’s ranked order correlation coefficients between characteristics and WHO-HPQ and *p*-values were calculated.

**Table 2 cancers-17-01742-t002:** Comparisons of the model fit indicators between the models.

*N* = 202									
Models ^a^	CMIN	Change ^b^	*p* ^c^	CMIN/df	CFI	GFI	AGFI	RMSEA	AIC
Model without any path constraints	3.11	NA	NA	0.389	1.000	0.994	0.977	<0.001	47.112
Model with path constraints									
PAM13 to cancer-related symptoms	5.11	2.00	0.16	0.567	1.000	0.974	0.954	<0.001	47.107
PAM13 to WHO-HPQ	3.26	0.15	0.70	0.362	1.000	0.978	0.961	<0.001	45.261
Cancer-related symptoms to WHO-HPQ	4.55	1.43	0.23	0.505	1.000	0.975	0.956	<0.001	46.545
All paths	7.02	3.91	0.27	0.638	1.000	0.971	0.954	<0.001	45.201

Note. AIC, Akaike Information Criterion; CMIN, chi-square statistics; CFI, Comparative Fit Index; df, degrees of freedom; GFI, Goodness-of-Fit Index; AGFI, Adjusted Goodness-of-Fit Index; PAM13, Patient Activation Measure-13; RMSEA, Root Mean Square Error of Approximation; WHO-HPQ, World Health Organization Health and Performance Questionnaire. ^a^ Equality constraints were placed between the low and high support groups. ^b^ Changes in chi-square statistics compared to the chi-square of the model without any path constraints. ^c^ *p*-values were calculated to indicate the significance of the chi-square change.

## Data Availability

The datasets generated and/or analyzed during the current study are not publicly available because the authors did not receive approval from the Ethics Committee of the principal investigator’s institution to share the collected data publicly. Inquiries for data access may be addressed to the corresponding author (Takafumi Soejima, soejimat@people.kobe-u.ac.jp).

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
