# Peer review of "The Relationship Between Patient Activation, Cancer-Related Symptoms, and Job Performance Among Young Adult Cancer Survivors with Low and High Workplace Support: A Cross-Sectional Observational Study"

_cancers, 2025, doi:10.3390/cancers17111742_

Round 1
Reviewer 1 Report
Comments and Suggestions for Authors
Overview: This study presents cross-sectional survey data examining the relationship between patient activation, cancer-related symptoms, and job performance among a sample of young adult cancer survivors in Japan. The level of workplace support was considered. Strengths include the focus on job performance and workplace support for young survivors, the use of structural equation modeling, the transparent reporting of statistical analyses and results, and the clear and thoughtful report of study limitations.
Major Comments:
- Cancer-related symptoms need to be defined earlier in the manuscript. It is not until the measures section that it becomes clear that these refer specifically to physical fatigue, depressive symptoms (which may or may not be related to cancer), and cognitive functioning.
- The response rate is quite low; were there any demographic or clinical characteristic differences between those who responded and those that didn’t? In other words, how representative of the population is the sample that responded?
- Given the cross-sectional nature of this study, it is possible that the relationships amongst variables occur in a different direction. Specifically, it’s possible that higher cancer-related symptoms reduce patient activation, which results in lower job performance. Please comment on alternative models and how future research might determine the direction of these effects.
Author Response
Comment 1: Overview: This study presents cross-sectional survey data examining the relationship between patient activation, cancer-related symptoms, and job performance among a sample of young adult cancer survivors in Japan. The level of workplace support was considered. Strengths include the focus on job performance and workplace support for young survivors, the use of structural equation modeling, the transparent reporting of statistical analyses and results, and the clear and thoughtful report of study limitations.
Response 1: Thank you for your comment. We provide the original comments from the reviewers and our responses to these comments below.
Comment 2: Cancer-related symptoms need to be defined earlier in the manuscript. It is not until the measures section that it becomes clear that these refer specifically to physical fatigue, depressive symptoms (which may or may not be related to cancer), and cognitive functioning.
Response 2: Physical fatigue, depression, and cognitive impairment are common symptoms that occur in cancer survivors regardless of cancer type, and persist for years after completing treatment. We defined these symptoms as cancer-related symptoms because our study included survivors with a variety of cancer types. Thus, we clarified this point in the Introduction and added references.
Introduction (Page 2, Line 9–12)
Original: Some studies found that cancer survivors who were employed were at higher risk of poor job performance than individuals without a cancer history. Cancer-related symptoms including physical fatigue, depression, and cognitive impairment were risk factors for impaired job performance after controlling age, gender, educational level, the time since diagnosis, chemotherapy, type of occupation, and job demands.
Revised: Moreover, some cancer survivors often experienced physical fatigue, depression, and cognitive impairment which were related to cancer and its therapy and persisted for years after treatment completion. These symptoms were also risk factors for impaired job performance.
Added references
6. Bower J.E. Cancer-related fatigue--mechanisms, risk factors, and treatments. Nat Rev Clin Oncol 2014, 11, 597-609. DOI: 10.1038/nrclinonc.2014.127
7. Smith H.R. Depression in cancer patients: Pathogenesis, implications and treatment (Review). Oncol Lett 2015, 9, 1509-1514. DOI: 10.3892/ol.2015.2944
8. Ahles T.A.; Root J.C. Cognitive Effects of Cancer and Cancer Treatments. Annu Rev Clin Psychol 2018, 14, 425-451. DOI: 10.1146/annurev-clinpsy-050817-084903
Comment 3: The response rate is quite low; were there any demographic or clinical characteristic differences between those who responded and those that didn’t? In other words, how representative of the population is the sample that responded?
Response 3: The response rate was low in the present study. Approximately half of the participants were diagnosed with female genital cancer. In addition, many of the participants had received surgery and may have had less severe complications than those who had received chemotherapy or radiation therapy. Thus, our study participants did not reflect the population of Japanese young adult cancer survivors. Thus, we have added this point into the Discussion.
Discussion (Page 12, Line 3–9)
Original: First, the response rate of our study was low, and our study participation may be biased toward younger adult cancer survivors in better health condition, who are able to com-plete the questionnaire. In addition, our study participants included a high proportion of female genital and breast cancer survivors, reflecting the proportion of cancer types among young adults in Japan. Thus, our findings had limited generalizability.
Revised: First, the response rate of our study was low. The present study also included a high proportion of female genital cancer survivors, which reflected the proportion of cancer types among young adults in Japan. In addition, many of the participants had received surgery and may have had less severe complications than those who had received chemotherapy or radiation therapy. Thus, our study participants may be biased toward younger adult cancer survivors in better health condition, and our findings had limited generalizability.
Comment 4: Given the cross-sectional nature of this study, it is possible that the relationships amongst variables occur in a different direction. Specifically, it’s possible that higher cancer-related symptoms reduce patient activation, which results in lower job performance. Please comment on alternative models and how future research might determine the direction of these effects.
Response 4: We are not able to confirm whether the model with the arrow from cancer-related symptoms to patient activation or the model with the arrow from patient activation to cancer-related symptoms are appropriate, because these are equivalence models in SEM. In addition, we are not able to analyze the non-recursive model with the two-way arrows between cancer-related symptoms and patient activation under the condition that cancer-related symptoms and patient activation affect job performance. However, as described in background (Page 2, Line 9–12), previous studies (e.g., the study by Hibbard et al. who conceptualized patient activation and developed the PAM-13) suggested the influence from patient activation to cancer-related symptoms (the arrow from patient activation to cancer-related symptoms). Thus, the arrow from patient activation to cancer-related symptoms was set in this study. However, the direction of this effect was not statistically tested, and this point is added into the Discussion.
Discussion (Page 12, Line 21–26)
Original: Further study using a longitudinal design is needed to identify this causal association.
Revised: We set the path from patient activation to cancer-related symptoms was set on the basis of previous studies, but the SEM was not able to confirm whether the arrow from patient activation to cancer-related symptoms or from cancer-related symptoms to patient activation was appropriate. Further study should clarify the temporality of relationships between patient activation, cancer-related symptoms, and job performance using a longitudinal design to identify this causal association.

Reviewer 2 Report
Comments and Suggestions for Authors
The relationship between patient activation, cancer-related 2 symptoms, and job performance among young adult cancer sur- 3 vivors with low and high workplace support: A cross-sectional 4 observational study
Comments:
- “Figure 1”
I can acknowledge “cancer-related symptoms” affect “job performance”. But why “patient activation” affect “cancer-related symptoms”. The association should be otherwise.
- “Participants who were diagnosed with cancer at less than 20 years of age (n = 138), were unemployed (n = 120), did not complete treatment (n = 216), and did not complete the WHO HPQ (n = 17) were excluded” – this para should be in the method’s section instead. Please add statement to explain the justification.
- “Table 1. Demographic and clinical characteristics of participants”
Split this table into two since currently the authors only show frequency with % (low vs high support) and not descriptive for WHO-HPQ
- Types of cancer probably is one of the confounders in the analysis. How the authors address this in Table 2 and Figure 2?
- “Table 3. Comparisons of the model fit indicators between the models” I am concern with RMSEA = 0.000. No errors at all?
Author Response
Comment 1: “Figure 1” I can acknowledge “cancer-related symptoms” affect “job performance”. But why “patient activation” affect “cancer-related symptoms”. The association should be otherwise.
Response 1: We are not able to confirm whether the model with the arrow from cancer-related symptoms to patient activation or the model with the arrow from patient activation to cancer-related symptoms are appropriate, because these are equivalence models in SEM. In addition, we are not able to analyze the non-recursive model with the two-way arrows between cancer-related symptoms and patient activation under the condition that cancer-related symptoms and patient activation affect job performance. However, as described in background (Page 2, Line 9–12), previous studies (e.g., the study by Hibbard et al. who conceptualized patient activation and developed the PAM-13) suggested the influence from patient activation to cancer-related symptoms (the arrow from patient activation to cancer-related symptoms). Thus, the arrow from patient activation to cancer-related symptoms was set in this study. However, the direction of this effect was not statistically tested, and this point is added into the Discussion.
Discussion (Page 12, Line 21–26)
Original: Further study using a longitudinal design is needed to identify this causal association.
Revised: We set the path from patient activation to cancer-related symptoms was set on the basis of previous studies, but the SEM was not able to confirm whether the arrow from patient activation to cancer-related symptoms or from cancer-related symptoms to patient activation was appropriate. Further study should clarify the temporality of relationships between patient activation, cancer-related symptoms, and job performance using a longitudinal design to identify this causal association.
Comment 2: “Participants who were diagnosed with cancer at less than 20 years of age (n = 138), were unemployed (n = 120), did not complete treatment (n = 216), and did not complete the WHO HPQ (n = 17) were excluded” – this para should be in the method’s section instead. Please add statement to explain the justification.
Response 2: We moved this sentence from the Results to the Methods. This study investigated job performance, which are measured only for workers, for young adult cancer survivors. Thus, we excluded from participants who were diagnosed with cancer at age 20 or older, who did not complete treatment (i.e., participants who were not young adult cancer survivors), who were unemployed, and who did not complete the WHO-HPQ (i.e., participants for whom we did not collect data on job performance). We added these points into the Introduction and Methods.
Introduction (Page 2, Line 12–16)
Original: A previous study suggested that, among young adult cancer survivors, who are com-monly defined as individuals diagnosed with adult-onset cancer and aged 20 to 39 years [14], optimal work outcomes were considered more important than among older adult cancer survivors because young adult cancer survivors were in the formative years of their careers [15,16]. Although there have been studies on work outcomes among older adult cancer survivors, less is known about this population in young adulthood. A further understanding of the work outcomes and these related factors among young adult cancer survivors will be essential for the future development of relevant services and resources for optimal work conditions.
Revised: Among young adult cancer survivors, who are commonly defined as individuals aged 20-39 who were diagnosed with cancer in adulthood and have completed treatment, a further understanding of the work outcomes and these related factors will be essential for the future development of relevant services and resources for optimal work conditions.
Methods (Page 3, Line 49–Page 4, Line 5)
Original: Participants who were interested in this study logged into the online survey system provided by Macromill Inc. on their own devices (tablets, smartphones, etc.). They were informed about this study and gave researchers their consent to participate on the online survey system.
Revised: Participants who were interested in this study logged into the online survey system provided by Macromill Inc. on their own devices (tablets, smartphones, etc.). Of 6,161 young adult cancer survivors, 693 returned the online questionnaires (response rate, 11.3%). This response rate was similar to that in a previous study. Participants who were diagnosed with cancer at less than 20 years of age (n = 138), were unemployed (n = 120), did not complete treatment (n = 216), and did not complete the WHO-HPQ (n = 17) were excluded to investigate job performance in employed survivors. Finally, we analyzed data from 202 young adult cancer survivors.
They were informed about this study and gave researchers their consent to participate on the online survey system.
Comment 3: “Table 1. Demographic and clinical characteristics of participants” Split this table into two since currently the authors only show frequency with % (low vs high support) and not descriptive for WHO-HPQ
Response 3: Table 1 indicated the frequency and percentage of participants in each category of variables. For example, total 7 participants (2 in the low support group and 5 in the high support group) were in the category of the 20-24 years old in Age. As the reviewer’s comment, however, Table 1 is not descriptive for the WHO-HPQ. Thus, we added the WHO-HPQ scores for each category of variables.
Comment 4: Types of cancer probably is one of the confounders in the analysis. How the authors address this in Table 2 and Figure 2?
Response 4: We compared the model including only the main variables (i.e., patient activation, cancer-related symptoms, and job performance) with the model including the main and adjusted variables with significant differences between low and high support group (i.e., type of employment, cancer sites, and period from diagnosis), using the indicators of model fit. The model including only the main variables indicated better fit than the model including the main and adjusted variables. This point was added in Methods.
Methods (Page 5, Line 29–36)
Original: … , and these variables were compared between the low and high support groups using the chi-squared test and Welch’s t-test. Furthermore, for the WHO-HPQ, correlation ratios (η2) with categorical variables and Spearman’s rank correlation coefficients (r) with continuous variables were calculated to explore the demographic and clinical characteristics that potentially confounded the association of job performance with patient activation and cancer-related symptoms.
Revised: Differences in categorical and continuous variables between the low and high support groups were assessed by the chi-squared test and Mann-Whitney U test, respectively. Furthermore, for the WHO-HPQ, correlation ratios (η2) with categorical variables and Spearman’s rank correlation coefficients (r) with continuous variables were calculated. The demographic and clinical characteristics with significant differences between low and high support group or significantly associated with WHO-HPQ were identified as potential confounders for the association of job performance with patient activation and cancer-related symptoms.
Methods (Page 5, Line 42–Page 6, Line 14)
Original: First, we developed the model without any path constraints on the relationship between patient activation, cancer-related symptoms, and job performance. Cancer-related symptoms were set as a latent variable consisting of physical fatigue, depression, and cognitive impairments. Second, the model fit indicators were compared between the model without any path constraints and the models with equality constraints of each path and all paths to assess whether each relationship between the main variables was moderated depending on workplace support.
Revised: First, we developed the model on the relationship between patient activation, cancer-related symptoms, and job performance.Cancer-related symptoms were set as a latent variable consisting of physical fatigue, depression, and cognitive impairments. Second, we compared the model including only the main variables (i.e., patient activation, cancer-related symptoms, and job performance) with the model including the main and adjusted variables with significant differences between low and high support group, using the indicators of model fit. Type of employment (1 = Temporary, 0 = Others), cancer sites (1 = Female genital cancer, 0 = Others), and period from diagnosis (1 = 5 years or more, 0 = Others) were entered as adjusted variables into the model. In the model including the main and adjusted variables, there was a significant association between type of employment and patient activation, but no other significant association between the main and adjusted variables. Thus, the model including the main variables and type of employment with the path from type of employment and patient activation was created. The model including only the main variables (CMIN/df = 0.39, CFI = 1.00, GFI = 0.99, AGFI = 0.98, and RMSEA = 0.00) indicated better fit than the model including the main variables and type of employment (CMIN/df = 1.25, CFI = 0.97, GFI = 0.94, AGFI = 0.90, and RMSEA = 0.04). We therefore used the multiple-group SEM model, including only patient activation, cancer-related symptoms, and job performance, in further analyses. Third, the model fit indicators were compared between the model without any path constraints and the models with equality constraints of each path and all paths to assess whether each relationship between the main variables was moderated depending on workplace support.
Comment 5: “Table 3. Comparisons of the model fit indicators between the models” I am concern with RMSEA = 0.000. No errors at all?
Response 5: The RMSEA value of 0.000 in Table 3 indicates that the RMSEA ranges from 0.001 to 0, not that the RMSEA is 0. We revised the RMSEA value in Table 3. Also, the closer RMSEA is to 0, the better the model is. A low RMSEA value have no problem (see the article by Schermelleh-Engel K et al. cited in this study).

Reviewer 3 Report
Comments and Suggestions for Authors
- Abstract sub-headings should be added.
- Introduction should be shortened.
- Hypothesis details and model should be moved to methods section.
- Please clarify how did you check the normality of the data that you have and decided which statistical analyses should be applied.
- I find that there is controversy in the statistical analysis as the authors used mean (sd) but applied Spearman test and Welch's test which are non-parametric tests.
Author Response
Comment 1: Abstract sub-headings should be added.
Response 1: We added the sub-headings into the Abstract. To our knowledge, however, the published article did not include sub-headings in the Abstract. Instruction for Authors in Cancers (https://www.mdpi.com/journal/cancers/instructions) stated “Systematic reviews and original research articles should have a structured abstract of around 250 words and contain the following headings: Background/Objectives, Methods, Results, and Conclusions.” Thus, we don't know if it is appropriate to include sub-headings.
Comment 2: Introduction should be shortened.
Response 2: In response to your Comment 3, we shortened the Introduction section, and, for example, we moved our hypotheses from the Introduction to the Methods.
Comment 3: Hypothesis details and model should be moved to methods section.
Response 3: According to your comment, we move our hypotheses from the Introduction to the Methods.
Methods (Page 5, Line 37–39)
Original: Multiple-group structural equation modeling (SEM) was … .
Revised: We hypothesized that, only among young adult cancer survivors with low workplace support, patient activation would be directly related to job performance, as well as indirectly related to job performance through cancer-related symptoms (Figure 1). Multiple-group structural equation modeling (SEM) was … .
Comment 4: Please clarify how did you check the normality of the data that you have and decided which statistical analyses should be applied.
Response 4: Especially, the CFS, K6, and EORTC QLQ-C30 scores were not a normal distribution (they are the right-skewed distribution). In the present study, we used a bootstrap method that allows robust estimation of parameters in structural equation modeling even if variables are not normally distributed. We added this point into the Methods.
Methods (Page 6, Line 23–26)
Original: Standardized path coefficients (β) with 95% confidence intervals (CIs) were estimated using a bootstrap analysis with 2,000 iterations and were tested by the bias-corrected bootstrap test.
Revised: Standardized path coefficients (β) with 95% confidence intervals (CIs) were estimated using a bootstrap analysis with 2,000 iterations and were tested by the bias-corrected bootstrap test to robustly estimate parameters in structural equation modeling with variables that are not normally distributed.
Comment 5: I find that there is controversy in the statistical analysis as the authors used mean (sd) but applied Spearman test and Welch's test which are non-parametric tests.
Response 5: Median and Interquartile ranges were calculated for continuous variables, and Mann-Whitney U test was used to compare continuous variables between low and high support groups. We revised the Methods and Table 1, and Table 1 and 2 were combined.
Methods (Page 5, Line 28–31)
Original: Frequencies, percentages, means, or standard deviations (SD) of demographic and clinical characteristics, patient activation, cancer-related symptoms, and job performance were calculated, and these variables were compared between the low and high support groups using the chi-squared test and Welch’s t-test
Revised: Frequencies and percentages for categorical variables and median with interquartile range (IQR) for continuous variables were calculated. Differences in categorical and continuous variables between the low and high support groups were assessed by the chi-squared test and Mann-Whitney U test, respectively.

Round 2
Reviewer 3 Report
Comments and Suggestions for Authors
No further comments.